# GENERATING IMAGES WITH 3D ANNOTATIONS USING DIFFUSION MODELS

**Wufei Ma**[1][*] **Qihao Liu**[1][*] **Jiahao Wang**[1][*] **Angtian Wang**[1]**, Xiaoding Yuan**[1]**,**
**Yi Zhang**[1]**, Zihao Xiao**[1]**, Guofeng Zhang**[1]**, Beijia Lu**[1]**, Ruxiao Duan**[1]**, Yongrui Qi**[1]**,**
**Adam Kortylewski**[2,3]**, Yaoyao Liu**[1][✉]**, Alan Yuille**[1]

[1]Johns Hopkins University     [2]University of Freiburg
[3]Max Planck Institute for Informatics, Saarland Informatics Campus

## ABSTRACT

Diffusion models have emerged as a powerful generative method, capable of producing stunning photo-realistic images from natural language descriptions. However, these models lack explicit control over the 3D structure in the generated images. Consequently, this hinders our ability to obtain detailed 3D annotations for the generated images or to craft instances with specific poses and distances. In this paper, we propose 3D Diffusion Style Transfer (3D-DST), which incorporates 3D geometry control into diffusion models. Our method exploits ControlNet, which extends diffusion models by using visual prompts in addition to text prompts. We generate images of the 3D objects taken from 3D shape repositories (e.g., ShapeNet and Objaverse), render them from a variety of poses and viewing directions, compute the edge maps of the rendered images, and use these edge maps as visual prompts to generate realistic images. With explicit 3D geometry control, we can easily change the 3D structures of the objects in the generated images and obtain ground-truth 3D annotations automatically. This allows us to improve a wide range of vision tasks, e.g., classification and 3D pose estimation, in both in-distribution (ID) and out-of-distribution (OOD) settings. We demonstrate the effectiveness of our method through extensive experiments on ImageNet-100/200, ImageNet-R, PASCAL3D+, ObjectNet3D, and OOD-CV. The results show that our method significantly outperforms existing methods, e.g., 3.8 percentage points on ImageNet-100 using DeiT-B. Our code is available at https://ccvl.jhu.edu/3D-DST/

## 1 INTRODUCTION

Understanding the underlying 3D world of 2D images is essential to numerous computer vision tasks. The utilization of 3D modeling opens up the possibility of addressing a significant portion of the variability inherent in natural images, which could potentially enhance the overall understanding and interpretation of images (Wu et al., 2020). For example, 3D-aware models show high robustness and generalization ability under occlusion or environmental changes (Liu et al., 2022a). However, it is expensive and time-consuming to obtain ground-truth 3D annotations for 2D images. This training data shortage becomes a main obstacle to training large-scale 3D-aware models.

Recently, diffusion models (Ho et al., 2020) have shown impressive performance in generating photo-realistic images, which can be used to solve the training data shortage. These models allow us to produce high-quality images from various conditional inputs, e.g., natural language descriptions, segmentation maps, and keypoints (Zhang et al., 2023). This facilitates generative data augmentation, e.g., He et al. (2023) use diffusion models to augment ImageNet (Deng et al., 2009) and significantly improve the classification results.

Despite their success, diffusion models still lack explicit control over the underlying 3D world during the generation process. As a result, they still face two challenges that hinder their use in augmenting data for 3D tasks. The first challenge is the inability to control the 3D properties of the object in

---

[*] Equal contribution.

[✉] Corresponding author: Yaoyao Liu (yyliu@cs.jhu.edu).

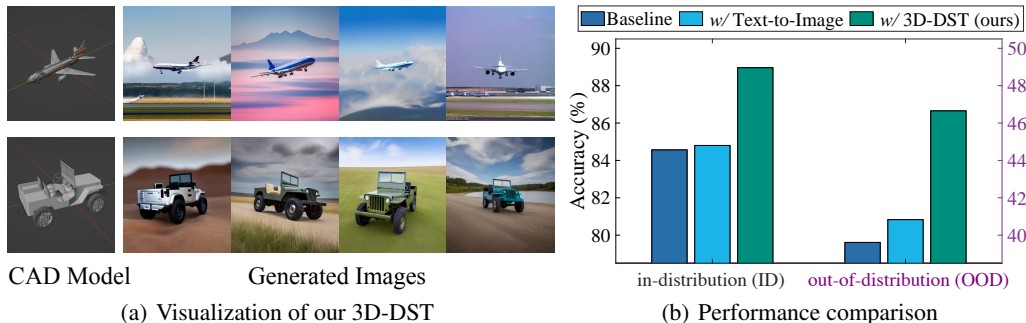

(a) Visualization of our 3D-DST          (b) Performance comparison

Figure 1: (a) **Visualization of our 3D-DST.** Our proposed solution, 3D-DST, leverages both 3D visual prompts and large language model (LLM) text prompts to generate diverse images from a CAD model. The use of 3D visual prompts enables explicit control over the 3D structure of the object within the generated images, such as varying 3D poses and distances. On the other hand, LLM text prompts facilitate the automatic generation of images with diverse backgrounds, weather conditions, and colors. (b) **Performance comparison.** Our model can be utilized to generate data to enhance both in-distribution (ID) and out-of-distribution (OOD) performance. We report results of DeiT-S on ImageNet-100 (Tian et al., 2020) and ImageNet-R (Hendrycks et al., 2021). "Text-to-Image" denotes using diffusion models without 3D control to augment the data (He et al., 2023).

the generated images, such as pose and distance. The second challenge is the difficulty in obtaining ground-truth 3D annotations of objects automatically.

To tackle the above challenges, we propose a simple yet effective framework, 3D Diffusion Style Transfer (3D-DST), which enables us to incorporate knowledge about 3D geometry structures. Our method exploits ControlNet (Zhang et al., 2023), which extends diffusion models by using visual prompts in addition to text prompts. We generate images of 3D objects taken from 3D shape repositories (e.g., ShapeNet (Chang et al., 2015) and Objaverse (Deitke et al., 2023b)), render them from a variety of viewing directions and distances, compute the edge maps of the rendered images, and use these edge maps as visual prompts to generate realistic images. With explicit 3D geometry control, we can easily change the 3D structures of the objects in the generated images and obtain corresponding 3D annotations automatically.

To enhance the diversity of the generated images, we apply the following strategies. Firstly, we vary the viewing directions in which the 3D objects are rendered. This generates a wide range of edge maps and allows us to produce multiple distinct images for each 3D object. Secondly, we introduce a novel prompting technique to improve diversity further. We input essential information about the 3D objects into large language models (LLM) (Touvron et al., 2023) to obtain reasonable descriptions, such as background and color. Then, we use these descriptions as the text prompts for diffusion models. This not only enables us to make the most of the vast potential of the diffusion models but also helps us to avoid generating images that are too similar. These two strategies allow us to generate diverse images that can be used to improve the out-of-distribution (OOD) robustness of AI models. As shown in Figure 1, our 3D-DST effectively generates images with a wide range of viewpoints, distances, colors, and backgrounds. These generated images have proven valuable in enhancing performance across both in-distribution (ID) and OOD scenarios.

Our 3D-DST allows us to transfer recently released large-scale 3D object datasets, e.g., Objaverse and Objaverse-XL, into 2D datasets enriched with comprehensive 3D annotations, e.g., 3D poses, key points, and depths. Objaverse is a repository of 800K CAD models, while Objaverse-XL (Deitke et al., 2023a) expands to more than 10 million 3D objects. With the capabilities offered by our 3D-DST, we are able to harness the potential of these vast datasets to enhance performance across various tasks, including 3D-aware classification, 3D pose estimation, OOD classification, and more.

We conducted extensive experiments to show the effectiveness of our method. Firstly, we show that our method can be directly used as a data augmentation method for classification. For example, our method improves the ImageNet-100 accuracy by 3.8 percentage points on DeiT-B (Touvron et al., 2021). Secondly, we demonstrate that our method is able to boost the performance of 3D-aware models. With automatically produced 3D annotations, pre-training on our generated images can improve the 3D pose estimation benchmark, PASCAL3D+ (Xiang et al., 2014), by 3.9 and 2.4 percentage points in ID and OOD settings, respectively (accuracy @$\frac{\pi}{18}$).

To summarise, we make three contributions:

- **Automatic generation of 3D annotations.** We propose a simple yet effective pipeline that allows us to add 3D conditional control to diffusion models. This enables us to acquire 3D annotations for the generated images through the rendering process.
- **Generating images with multiple viewpoints.** Our method generates images from diverse viewpoints, including those that are rarely encountered in typical scenarios. They can be used as training to improve the model's robustness in the OOD setting.
- **Diverse text prompts by LLM.** We propose a novel strategy for text prompt generation using LLM. This strategy effectively prevents the generation of redundant or similar images.

## 2 RELATED WORK

**Synthetic data augmentation.** Synthetic data has gained significant attention in generating labeled data for vision tasks that require extensive annotations (Rombach et al., 2022; Zhang et al., 2023). The synthetic data augmentation methods can be categorized into two groups: 1) *2D-based methods* employ recent generative models like GANs and diffusion models to create photo-realistic images (Baranchuk et al., 2022; Liu et al., 2021a; Dosovitskiy et al., 2015; Sun et al., 2021), and other dense prediction tasks. Despite their effectiveness, these methods lack 3D structures, making it challenging to acquire 3D annotations for the generated images. 2) *3D-based methods* leverage simulation environments with physically realistic engines to render 3D models and generate images (Greff et al., 2022; Zheng et al., 2020). However, the generated images' diversity is limited as they heavily rely on the existing textures of the 3D models. In our work, we investigate the integration of 3D control into diffusion models. This allows us to generate diverse images using the appearance produced by diffusion models while obtaining 3D annotations through the 3D structure conditions.

**Diffusion models** operate by incrementally degrading the data through introducing Gaussian noise gradually, and subsequently learn to restore the data by reversing this noise infusion process (Ho et al., 2020; Singer et al., 2023; Villegas et al., 2023). They have shown remarkable success in generating high-resolution photo-realistic images from various conditional inputs, e.g., natural language descriptions, segmentation maps, and keypoints (Ho et al., 2020; 2022; Zhang et al., 2023). Recently, text-to-image diffusion models have also been used to augment training data. Trabucco et al. (2023) investigate various strategies for augmenting individual images with the help of a pre-trained diffusion model, showcasing considerable enhancements in few-shot learning scenarios. Azizi et al. (2023) present evidence that the usage of class names as text prompts can guide the diffusion models to generate images that subsequently improve performance in ImageNet classification tasks. Despite their advancements, diffusion models still face limitations in explicitly controlling the 3D structure of the images they generate. Our research contributes to overcoming this challenge by integrating 3D geometry conditional inputs into diffusion models. This enhancement empowers us with precise control over the 3D structure of the object within the produced image, and facilitates effortless acquisition of 3D annotations, such as 3D pose key points.

**Large language models (LLM).** The field of natural language processing has witnessed a transformative shift in recent years, spurred by the advent of large language models such as PaLM (Chowdhery et al., 2023), and LLaMA (Touvron et al., 2023). These large language models have showcased remarkable proficiency in zero-shot and few-shot tasks, as well as more intricate assignments like mathematical problem-solving and commonsense reasoning. Their impressive performance can be attributed to the extensive corpora they are trained on and the intensive computational resources dedicated to their training. In our study, we utilize LLaMA to enhance the quality of text prompts. By automatically generating descriptive prompts pertaining to backgrounds, color, and weather conditions, we successfully improve the diversity of our generated images.

## 3 3D DIFFUSION STYLE TRANSFER (3D-DST)

As illustrated in Figure 2, our 3D-DST enhances diffusion models by incorporating both 3D visual prompts and diverse text prompts. In Section 3.1, we describe how existing diffusion models incorporate 2D visual and simple text conditions for generative data augmentation. In Section 3.2, we introduce how to produce visual prompts based on 3D geometry conditions by graphics-based rendering. In Section 3.3, we show how to create text prompts with LLM to enhance diversity. Algorithm 1 summarizes how to generate images with 3D annotations using our 3D-DST.

### 3.1 Background: Diffusion Models with Text and 2D Visual Prompts

Diffusion models (Ho et al., 2020; Rombach et al., 2022) achieve great success in conditional image generation using text and 2D visual prompts. In the following, we first introduce the original diffusion models without conditional inputs. Then, we show how to add text prompts by augmenting the underlying U-Net backbone of the diffusion models with a cross-attention mechanism. Moreover, we elaborate on how to integrate 2D visual prompts using ControlNet.

**Diffusion model pipeline.** Diffusion models learn to generate images by learning a sequence of denoising U-Nets. Starting from a random noise $z_T$, the denoising process is as follows,

$$z_{t-1} = \epsilon(z_t, t), \quad t = T \ldots 1, \tag{1}$$

where $\epsilon(z_t, t)$ denotes the denoising U-Nets. $z_{t-1}$ is a denoise version of the input $z_t$. In the final step, $z_0$ is input to a pre-trained decoder $\mathcal{D}$ to get the generated image $I_{\text{final}}$, i.e., $I_{\text{final}} = \mathcal{D}(z_0)$.

**Adding text prompts via cross-attention.** To enable text-conditioning and generate images adhered to specific content requirements, previous works (Zhang et al., 2023; Azizi et al., 2023; Rombach et al., 2022) incorporate text prompts into diffusion models using the cross-attention mechanism (Vaswani et al., 2017), following Rombach et al. (2022). Specifically, text prompts $\mathcal{T}$ are preprocessed with a pre-trained text encoder $\theta$, which are then integrated to intermediate layers of the U-Net $\epsilon$ via cross-attention layers:

$$\phi_i = \text{softmax}\left(\frac{QK^M}{\sqrt{|\phi_{i-1}|}}\right) \cdot V, \quad Q = W_Q^{(i)} \cdot \phi_{i-1}, \quad K = W_K^{(i)} \cdot \theta(\mathcal{T}), \quad V = W_V^{(i)} \cdot \theta(\mathcal{T}), \tag{2}$$

where $\phi_i$ denotes the $i$-th layer intermediate representation of the U-Net $\epsilon$, and $\phi_i = z_t$. $W_Q^{(i)}, W_K^{(i)}$, and $W_V^{(i)}$ are learnable projection matrices. $M$ is a hyperparameter of the cross-attention.

**Adding 2D visual prompts via ControlNet.** To integrate 2D visual prompts into diffusion models, we utilize the ControlNet (Zhang et al., 2023) architecture, which incorporates 2D visual prompts without retraining the entire diffusion model. During Step $t$ of the denoising process, the 2D visual prompts $\mathcal{E}_{\text{2D}}$ are added to the latent variable $z_t$, and the resulting sum is inputted into the ControlNet. Then, the outputs of the ControlNet will be integrated into the denoising U-Net $\epsilon$, i.e.,

$$z_{t-1} = \epsilon(z_t, \mathcal{T}, \text{ControlNet}(z_t + \mathcal{E}_{\text{2D}})). \tag{3}$$

**Limitations of existing methods.** Although existing approaches have achieved notable success, they still encounter the following challenges. Firstly, these methods lack 3D geometry control, making it arduous to explicitly modify the 3D structure of objects in generated images and acquire corresponding 3D annotations, such as 3D pose key points. Secondly, existing methods frequently depend on simple text prompts to guide the image generation process when augmenting datasets with diffusion models. Consequently, it is difficult to create images for OOD scenarios, which is very important for training robust models.

In order to address these challenges, in the following, we demonstrate the generation of 3D visual prompts through graphics-based rendering and augment the text prompts with LLM, effectively enhancing the diversity of the generated images.

### 3.2 3D Visual Prompts by Graphics-based Rendering

In order to effectively integrate 3D geometry control into diffusion models, it is essential to ensure that the visual prompts meet the following two specific requirements.

Firstly, the visual prompts must encompass sufficient information to depict the 3D geometry structure of the objects accurately. Without this information, it would be challenging to achieve explicit control over the 3D structure of the objects (e.g., shape and pose) in the generated images, consequently hindering the generation of corresponding 3D annotations.

Secondly, the visual prompts should be compact and concise, enabling diffusion models to comprehend and effectively process them. Utilizing overly complex visual prompts, such as the vertices

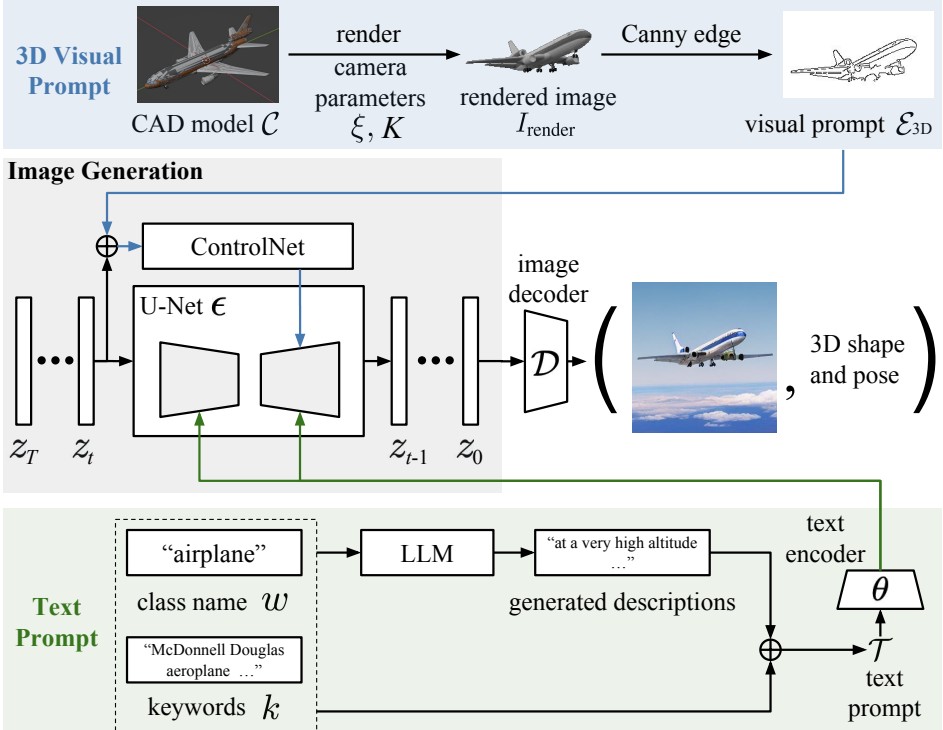

Figure 2: Our 3D-DST comprises three essential steps. (1) **3D visual prompt generation**. We generate images of 3D objects taken from a 3D shape repository (e.g., ShapeNet and Objaverse), render them from a variety of viewpoints and distances, compute the edge maps of the rendered images, and use these edge maps as 3D visual prompts. (2) **Text prompt generation**. Our approach involves combining the class names of objects with the associated tags or keywords of the CAD models. This combined information forms the initial text prompts. Then, we enhance these prompts by incorporating the descriptions generated by LLaMA. (3) **Image generation.** We generate photo-realistic images with 3D visual and text prompts using Stable Diffusion and ControlNet.

and meshes of a CAD model, would prove impractical and unfeasible for the diffusion models to handle efficiently. Therefore, it is crucial to strike a balance and employ visual prompts that are both informative and manageable within the diffusion model framework.

To satisfy the aforementioned requirements, our proposed approach involves generating visual prompts through graphics-based rendering. We leverage 3D Computer Aided Design (CAD) models, which can be easily obtained from existing 3D shape repositories such as ShapeNet, Objaverse, and Objaverse-XL. Then, we render the CAD models from diverse viewpoints and distances, compute edge maps based on the rendered images, and utilize these edge maps as visual prompts. This approach enables us to encapsulate the necessary information regarding the 3D geometry structure of the objects. The rendered images provide essential details such as the viewing directions and distances, facilitating explicit control over the 3D structure during the generation of images. By extracting edge maps, we create compact visual prompts that are suitable for integration within diffusion models. In the following, we will elaborate on the detailed steps.

**Graphics-based rendering.** We render CAD models with various viewing directions and distances and then obtain the rendered sketch images. Given a CAD model $\mathcal{C}$, we generate the rendered sketch images $I_{\text{render}}$ as follows,

$$I_{\text{render}} = R(\mathcal{C}, \xi, K), \tag{4}$$

where $R$ denotes an off-the-shelf renderer, i.e. (Community, 2018), $K$ is the camera intrinsic matrix, $\xi$ represents a camera extrinsic matrix computed from a randomized viewing direction, and distance follows a predefined distribution. In practice, we use the Perspective camera with focal length $f = 35$mm to render the images. Complex backgrounds are not necessary here because the rendered sketch images are used to extract edge maps.

**Edge map computation.** Next, we extract edge maps from the rendered sketch images. The primary objective of this process is to enhance the compactness and conciseness of the rendered images while preserving the underlying 3D geometry structure. To achieve this, we apply classical edge detection methods, e.g., Canny edge (Canny, 1986). Given a rendered sketch images $I_{render}$, we obtain the edge map $\mathcal{E}_{3D}$ as follows,

$$\mathcal{E}_{3D} = \text{CannyEdge}(I_{render}). \tag{5}$$

### 3.3 DIVERSE TEXT PROMPTS BY LLM

Text prompts play a crucial role in guiding diffusion models to generate diverse images. However, existing diffusion-model-based data augmentation methods often rely on overly simplistic text prompts. For example, Azizi et al. (2023) utilize class names directly as text prompts, while Zhang et al. (2023) employ a default prompt such as "a high-quality, detailed, and professional image." These methods fail to fully harness the rich appearance information stored within diffusion models.

To address the above issue, we present a novel strategy for text prompt generation. Our approach forms the initial text prompts using the class names of objects with the associated tags or keywords of the CAD models. Then, we enhance these prompts by incorporating the descriptions generated by LLM. The final text prompts $\mathcal{T}$ of a CAD model $\mathcal{C}$ are created as follows,

$$\mathcal{T} = \{t, \ w, \ k, \ \text{LLM}(t, w, k)\}, \tag{6}$$

---

**Algorithm 1** Generating images using our 3D-DST

1: **Input:** 3D shape repository $\{\mathcal{C}, w, k\}$.
2: **Output:** Images with 3D annotations $\{I_{final}, y_{3D}\}$.
3: **for** iterations **do**
4:   Get a CAD model $\mathcal{C}$ from the 3D shape repository;
5:   Load the class name $w$ and keyword $k$ of $\mathcal{C}$;
6:   Load the camera intrinsic matrix $K$;
7:   Generate randomized camera extrinsic matrices $\{\xi\}$;
8:   **for** different camera extrinsic matrix $\xi$ **do**
9:     // **3D visual prompt generation**
10:     Render sketch image $I_{render}$ using $\xi$ and $K$ by Eq. 4;
11:     Produce 3D annotations $y_{3D}$ of $I_{render}$;
12:     Compute edge map $\mathcal{E}_{3D}$ using Canny Edge by Eq. 5 ;
13:     // **Text prompt generation**
14:     Create text prompts $\mathcal{T}$ using $w$ and $k$ by Eq. 6;
15:     // **Image generation**
16:     Initialize a random noise $z_T$;
17:     **for** $t$ in $T, \cdots, 1$ **do**
18:       Denoise $z_t$ to $z_{t-1}$ using $\mathcal{T}$ and $\mathcal{E}_{3D}$ by Eq. 3;
19:     Generate images $I_{final} = \mathcal{D}(z_0)$.
20: Collect the generated dataset $\{I_{final}, y_{3D}\}$.

---

where $t$ represents a prompt template, such as "a photo of $\cdots$". $w$ corresponds to the class name of the CAD model $\mathcal{C}$, while $k$ denotes the tags or keywords associated with $\mathcal{C}$ in the 3D shape repository (e.g., Objaverse-XL). LLM is a large language model capable of generating rich and coherent descriptions (e.g., backgrounds, colors) when provided with the initial text prompts $(t, w, k)$.

## 4 EXPERIMENTS

### 4.1 IN-DISTRIBUTION (ID) AND OUT-OF-DISTRIBUTION (OOD) IMAGE CLASSIFICATION

**Datasets.** We consider three datasets: ImageNet-100, ImageNet-200, and ImageNet-R. The first two datasets are ID datasets, while the third one is an OOD dataset. All datasets are derived from ImageNet (Russakovsky et al., 2015), where ImageNet-100 (Tian et al., 2020; Liu et al., 2023; 2024) and ImageNet-200 are 100-class and 200-class subsets from random sampling. ImageNet-R (Hendrycks et al., 2021) contains $30,000$ images with various artistic renditions of 200 classes of the original ImageNet. It is widely used to evaluate OOD performance.

**Implementation details.** *Network architectures.* We use representative network architectures, i.e., DeiT (Touvron et al., 2021), ConvNeXt (Liu et al., 2022b), and Swin Transformer (Swin) (Liu et al., 2021b). For DeiT and ConvNeXt, we follow the official codebases and experimented on tiny (DeiT only), small, and base settings. No distillation is used for DeiT. For Swin Transformer and MAE, we experimented on the small setting following the training strategies released in the official implementations. *Data generation.* We collect around 30 CAD models from ShapeNet and Objaverse for each object class. Then, we run our 3D-DST and generate $2,500$ images for each class.

| Methods | In-distribution (ID) | | | | | | |
|---|---|---|---|---|---|---|---|
| | DeiT-Ti | DeiT-S | DeiT-B | ConvNeXt-S | ConvNeXt-B | Swin-S | MAE-S |
| Baseline | 83.84 | 84.56 | 84.28 | 91.34 | 91.50 | 90.78 | 88.69 |
| *w /* Text2Img (2023) | 84.58 (↑0.74) | 84.80 (↑0.24) | 84.10 (↓0.18) | 91.26 (↓0.08) | 91.28 (↓0.22) | 90.42 (↓0.36) | 88.78 (↑0.09) |
| *w /* 3D-DST (ours) | **87.36** (↑3.52) | **88.96** (↑4.40) | **88.11** (↑3.83) | **92.18** (↑0.84) | **92.44** (↑0.94) | **91.50** (↑0.72) | **90.47** (↑1.78) |

| Methods | Out-of-distribution (OOD) | | | | | | |
|---|---|---|---|---|---|---|---|
| | DeiT-Ti | DeiT-S | DeiT-B | ConvNeXt-S | ConvNeXt-B | Swin-S | MAE-S |
| Baseline | 49.96 | 49.61 | 50.53 | 67.19 | 66.40 | 54.99 | 65.18 |
| *w /* Text2Img (2023) | 52.58 (↑2.62) | 50.83 (↑1.22) | 49.61 (↓0.92) | 67.50 (↑0.31) | 67.15 (↑0.75) | 56.34 (↑1.35) | 67.28 (↑2.10) |
| *w /* 3D-DST (ours) | **56.12** (↑6.16) | **56.65** (↑7.04) | **56.74** (↑6.21) | **69.69** (↑2.50) | **69.21** (↑2.81) | **59.97** (↑4.98) | **68.42** (↑3.24) |

Table 1: Image classification accuracy (%) on ImageNet-100 (ID) and ImageNet-R (OOD) using representative network architectures, ResNet and ViT. We compare the performances when models are (1) trained purely on the target dataset, (2) pre-trained on Text2Img (He et al., 2023) data, which does not have 3D control, and then finetuned on the target dataset, (3) pre-trained on 3D-DST data, and finetuned on the target dataset. Experiments show that our 3D-DST data can help boost the classification accuracy of both models on both ID and OOD cases by a large margin.

**ID results on ImageNet-**100 **and ImageNet-**200. In Table 1, we show the ID classification results on ImageNet-100. "Baseline" (Line 1) is to train models purely on ImageNet-100 for 600 epochs. "*w/* Text2Img" (Line 2) is to pre-train the models on the images generated by a text-to-image (Text2Img) model (He et al., 2023) without 3D control for 300 epochs and finetune on ImageNet-100 for 300 epochs. "*w/* 3D-DST" (Line 3) is to pre-train the models on our 3D-DST synthetic data first for 300 epochs and finetune on

| Methods | Accuracy (%) |
|---|---|
| Baseline | 81.50 |
| *w /* Text2Img (2023) | 83.45 (↑1.95) |
| *w /* 3D-DST (ours) | **84.81** (↑3.31) |

Table 2: Image classification accuracy on ImageNet-200 using DeiT-S.

ImageNet-100 for 300 epochs. For MAE in various settings, we follow the official implementation and pretrain the model for 800 epochs and then finetune on ImageNet-100 for 100 epochs. Comparing results on Line 3 and Line 1, we can observe with the help of 3D-DST data, the Top-1 accuracies increase by more than 3.50 percentage points for DeiT models and an average of 1.07 percentage points for other models with an accuracy of more than 90 percentage points. As a comparison, pretraining on Text2Img (Line 2) yields mixed results with no evident improvements. In Table 2, we provide the results on ImageNet-200. We can see that using our 3D-DST can still achieve significant improvements, i.e., 3.31 percentage points compared to the baseline.

**OOD results on ImageNet-R.** Table 1 also demonstrate the OOD classification results on ImageNet-R. For "Baseline", "*w/* Text2Img", and "*w/* 3D-DST", we apply the similar settings as the ID experiments. Comparing results on Line 2 with Line 1, we can observe that Text2Img improves the performance on ImageNet-R by a small marge with an average of 1.06 percentage points, while pretraining models on our 3D-DST data bring a consistent and significant improvement across all models, with an average of 4.70 percentage points.

**Choice of different data generation methods.** To demonstrate the advantages of our proposed 3D-DST over, we consider the following data generalization baselines and present the quantitative results in Table 6. (i) Diffusion-based generation conditioned on edges obtained from ImageNet images. (ii) Rendering images with random 2D image backgrounds. (iii) Rendering images with random 3D environment backgrounds. Please refer to Section D where we discuss the detailed implementations and present qualitative examples from each method.

**Ablation results.** In Table 3, we provide the ablation results. Line 1 shows the baseline results without pre-training. Line 2 and Line 3 report the results that pre-train on 3D-DST data with and without LLM prompts. As we can see, using LLM can significantly improve the performance on both in-distribution and out-of-distribution data. Note that the LLM prompts are generated with our novel generation strategy introduced in Section 3.3, effectively improving the diversity of the produced prompts. As a result, LLM prompts achieve larger improvements in OOD compared to ID.

## 4.2 ROBUST CATEGORY-LEVEL 3D POSE ESTIMATION

**Datasets and evaluations.** *Datasets*. We consider three datasets for 3D pose estimation: PASCAL3D+, ObjectNet3D, and OOD-CV. The PASCAL3D+ dataset (Xiang et al., 2014) contains

| No. | 3D-DST (pre-train) | LLM | In-distribution (ID) | | | Out-of-distribution (OOD) | | |
|-----|-----|-----|--------|--------|------------|---------|--------|------------|
| | | | DeiT-Ti | DeiT-S | ConvNeXt-S | DeiT-Ti | DeiT-S | ConvNeXt-S |
| 1 | | | 83.84 | 84.56 | 91.34 | 49.96 | 49.61 | 67.19 |
| 2 | ✓ | | 86.52 (↑2.68) | 87.82 (↑3.26) | 91.96 (↑0.62) | 54.51 (↑4.55) | 55.95 (↑6.34) | 68.29 (↑1.10) |
| 3 | ✓ | ✓ | **87.36** (↑3.52) | **88.96** (↑4.40) | **92.18** (↑0.84) | **56.12** (↑6.16) | **56.65** (↑7.04) | **69.69** (↑2.50) |

Table 3: Ablation Study (%) in the ID and OOD settings. Numbers in brackets show rise/fall compared to the baseline. For ID, we use ImageNet-100. For OOD, we use ImageNet-R. Line 1 shows the baseline results without pre-training. Lines 2 and 3 report the results that pre-train on 3D-DST data with and without LLM prompts.

| Methods | In-distribution (ID) | | Out-of-distribution (OOD) | |
|---------|------------|-------------|------------|-------------|
| | Acc@$\pi/6$ | Acc@$\pi/18$ | Acc@$\pi/6$ | Acc@$\pi/18$ |
| ResNet | 82.33 | 52.60 | 50.38 | 23.38 |
| ResNet *w/* AugMix (2020) | 82.72 (↑0.39) | 53.89 (↑1.29) | 51.77 (↑1.39) | 24.57 (↑1.19) |
| ResNet *w/* 3D-DST (ours) | 84.22 (↑1.89) | 56.52 (↑3.92) | 52.75 (↑2.37) | 25.70 (↑2.32) |
| NeMo (2021) | 82.23 | 57.12 | 55.31 | 26.57 |
| NeMo *w/* AugMix (2020) | 83.11 (↑0.88) | 58.22 (↑1.10) | 56.38 (↑1.07) | 26.63 (↑0.06) |
| NeMo *w/* 3D-DST (ours) | 85.70 (↑3.47) | 62.51 (↑5.39) | 58.81 (↑3.50) | 26.44 (↓0.13) |

Table 4: Robust 3D pose estimation on ID (PASCAL3D+ & ObjectNet3D (Xiang et al., 2016; 2014)) and OOD (OOD-CV (Zhao et al., 2022)). We experiment with a classification-based pose estimation method, ResNet, and a 3D compositional model, NeMo (Wang et al., 2021). Experimental results demonstrate that our DST synthetic data can effectively improve 3D pose estimation performance on both ID and OOD benchmarks.

$11,045$ training images and $10,812$ validation images with category and object pose annotations. The OOD-CV dataset (Zhao et al., 2022; 2023) includes OOD examples from PASCAL3D+ and is a benchmark to evaluate OOD robustness to individual nuisance factors, including pose, shape, appearance, context, and weather. The ObjectNet3D dataset is another 3D pose estimation benchmark that contains 100 categories with $17,101$ training samples and $19,604$ testing samples. Following Wang et al. (2021); Zhao et al. (2022), we evaluate pose estimation models on 10 categories from PASCAL3D+ and a subset of ObjectNet3D with 10 categories. ***Evaluations.*** Following Zhou et al. (2018); Wang et al. (2021), we measure the 3D pose prediction with the pose estimation error between the predicted rotation matrix and the ground truth rotation matrix $\Delta(\mathbf{R}_{\text{pred}}, \mathbf{R}_{\text{gt}}) = \frac{\|\log m(\mathbf{R}_{\text{pred}}^\top \mathbf{R}_{\text{gt}})\|_F}{\sqrt{2}}$. We report results under thresholds $\frac{\pi}{6}$ and $\frac{\pi}{18}$, following Zhou et al. (2018); Wang et al. (2021).

**ID Results on PASCAL3D+ and ObjectNet3D.** In Table 4, we show the ID pose estimation results on PASCAL3D+ and ObjectNet3D. The first block (Lines 1-3) shows ResNet results, i.e., extending a ResNet model with a pose classification head (Zhou et al., 2018). The second block (Lines 4-5) shows the results based on the state-of-the-art 3D pose estimation method, NeMo (Wang et al., 2021). "*w/* 3D-DST" denotes using the models pre-trained on 3D-DST. We also pre-train the model with a strong data augmentation method, AugMix (Hendrycks et al., 2020), which mixes augmented images and enforces consistent embeddings of the augmented images. "*w/* AugMix" shows its results. We can see that using the model pre-trained on our synthetic data can effectively improve the 3D pose estimation results. For example, comparing Line 3 with Line 1, we can observe that "*w/* 3D-DST" improves the 3D pose estimation results by $1.89$ and $3.47$ percentage points with a threshold of $\frac{\pi}{6}$ and by $3.92$ and $5.39$ percentage points for $\frac{\pi}{18}$, on ResNet. As a comparison, "*w/* AugMix" brings limited improvements compared to our 3D-DST when evaluated on the ID test data.

**OOD Results on on OOD-CV.** We also provide OOD pose estimation results on the OOD-CV dataset. For the baseline, "*w/* AugMix", and "*w/* 3D-DST", we apply the similar settings as the ID experiments. Comparing Line 3 with Line 1, our 3D-DST data can effectively improve the model's performance on OOD data, e.g., with a gap of $2.37$ and $3.50$ percentage points for $\frac{\pi}{6}$. This shows that our approach with 3D-DST can introduce diverse data not present in the training data and improve the robustness of models to various domain shifts.

**Real vs. Synthetic (3D-DST)**    **Diverse Appearances**

Figure 3: **Left:** Visualizations of 3D-DST synthetic data for image classification and 3D pose estimation. **Right:** Objects with various appearances can be generated from only a limited number of CAD models by conditioning on diverse textual prompts.

### 4.3 3D OBJECT DETECTION

To show the flexibility of our 3D-DST approach and its potential for a wide range of 3D tasks, we present qualitative and quantitative results on 3D object detection from scenes in ARKitScenes dataset (Baruch et al., 2021; Brazil et al., 2023). We present experimental settings in Section E and show that with 3D-DST we can effectively improve 3D object detection on both AP2D and AP3D metrics. This demonstrates that our 3D-DST is also applicable to complex scenes with multiple objects from multiple categories.

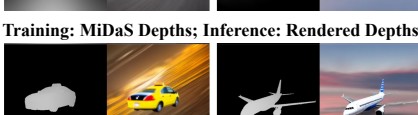

### 4.4 VISUALIZATIONS

Figure 3 shows the visualized samples generated by our 3D-DST. We can observe that 3D-DST can generate realistic images with diverse appearances. In Figure 4, we provide the comparisons among three types of 3D controls, edge maps, MiDaS (Ranftl et al., 2020) predicted depths, and rendered depths. Results show that using edge maps as control gives the best outputs visually, while MiDaS depths and rendered depths demonstrate reasonable results but are limited in the realism and clarity of the foreground object or the background scene.

Figure 4: Qualitative examples of using different types of 3D control. We experimented with three different types of 3D control: edge maps (top), MiDaS predicted depth (middle), and Blender rendered depth (bottom), using the same 3D model and text prompts. Qualitative results show that using edge maps as 3D control gives overall better outputs.

### 4.5 ANALYSES OF FAILURE CASES AND LIMITATIONS

We conduct a thorough analysis of the failure cases by our generation pipeline by collecting inputs from human evaluators. We further present a *K-fold consistency filter* (KCF) that can automatically detect and remove failed images. By analyzing the failure modes of our generation pipeline, we identify a limitation of our model to be the images with uncommon viewpoints (*e.g.*, looking at cars from below). Please refer to Section F for detailed results and discussions.

## 5 CONCLUSION

In this work, we introduce a simple yet effective framework, 3D-DST, which incorporates 3D geometry control into diffusion models. This empowers us with explicit control over the 3D structure of the objects in the generated images. As a result, we can conveniently acquire ground-truth 3D annotations for the produced 2D images. To boost the diversity of the images, we adjust 3D poses and distances and employ LLM for creating dynamic text prompts. Our empirical results reveal that the images generated by our method can significantly enhance performance across a range of vision tasks, including classification and 3D pose estimation, in both ID and OOD settings.

ACKNOWLEDGEMENTS

Alan Yuille acknowledges the Army Research Laboratory award W911NF2320008 and ONR N00014-21-1-2812. Adam Kortylewski acknowledges support for his Emmy Noether Research Group funded by the German Science Foundation (DFG) under Grant No. 468670075.

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

## A  ADDITIONAL RESULTS

**Qualitative examples of LLM prompts.**    To demonstrate how diverse prompts from LLM can help to generate diverse outputs in textures and backgrounds, we show some qualitative comparisons in Figure 6 between 3D-DST data with and without diverse prompts from LLM (when not using LLM, we only use the class names from ImageNet and descriptions from the CAD models as simple prompts).

**Error bar.**    It would be too computationally expensive to report the error bars for all the experiments. However, we compute the error bar for a main experiment in the paper, i.e., the top-1 classification accuracy of the three models trained on ImageNet-100 + 3D-DST, by only changing the random seeds and keeping all the other settings the same. The standard deviations of the classification accuracy of DeiT-S, ConvNeXt-S are $0.12\%$, $0.06\%$, respectively.

**Generating balanced data from 3D controllability.**    Another advantage of our 3D-DST dataset is to generate balanced data with respect to 3D viewpoint, object 2D location, or size, thanks to the 3D controllability of our generation pipeline. To visualize the 3D viewpoint distribution in different synthetic datasets, we randomly select 200 car/bus image samples from Text2Img and our 3D-DST, and run a state-of-the-art 6D pose estimation model to obtain the 3D viewpoint, NeMo6D (Ma et al., 2022). Then, we manually go over the estimated poses to remove samples with poorly estimated poses and obtain 120 well-annotated samples for each dataset. We visualize the distribution of the azimuth angles for 120 objects in Text2Img (left) and 3D-DST (right). Synthetic data in Text2Img show strong pose biases propagated from the image diffusion model, while our 3D-DST dataset alleviates this issue by introducing explicit 3D controls.

## B  MODEL TRAINING

**Image classification.**    We follow the implementation from the DeiT Touvron et al. (2021) codebase for all models. During training, the input image size is $224 \times 224$, the batch size is set to $512$, and we use AdamW Loshchilov & Hutter (2019) as the optimizer. The initial learning rate is $1e - 4$ and we use a cosine scheduler for learning rate decay. We use Rand-Augment Cubuk et al. (2020), random erasing Zhong et al. (2020), Mixup Zhang et al. (2018) and Cutmix Yun et al. (2019) for data augmentation. We use two NVIDIA Quadro RTX 8000 GPUs for each training. For other settings, we refer to Touvron et al. (2021).

**3D pose estimation.**    For the classification-based model, ResNet50, we use the released implementation from Zhou et al. (2018) and train the model for 90 epochs with a learning rate of 0.01. For NeMo, we adopt the publicly released code Wang et al. (2021) and trained the NeMo models for 800 epochs on both the synthetic and real data. Each NeMo model is trained on four NVIDIA RTX A5000 for 10 hours.

## C  MORE DETAILS ABOUT GENERATING 3D-DST SYNTHETIC DATA

The 100 classes we used in main classification experiments (Table 1) follow the 100 classes in (Tian et al., 2020). To demonstrate the scalability of our approach, we randomly select another 100 classes and report 200-class classification results in Table 2. We collect 30 CAD models for each class from the ShapeNet dataset (Chang et al., 2015) and the Objeverse dataset (Deitke et al., 2023b), and 2,500 images were rendered for each class. We sample the object viewpoint with a uniform distribution over the azimuth angle and Gaussian distributions over the elevation and theta angles. The viewpoint sampling rules are detailed in Table 5. For pose estimation, we estimate the mean and variance of the object viewpoints from the training data of PASCAL3D+ (Xiang et al., 2014) and ObjectNet3D (Xiang et al., 2016) and sample from the fitted Gaussian distributions.

## D  ABLATION STUDY EXPERIMENTS ON DATA GENERATION

Besides the results in Table 1 and Table 4, we consider the following ablation study experiments on different data generation methods. In the following we introduce different data generation procedures.

Sample images for each data generation approach are shown in Figure 8. Experimental results are presented in Table 6.

**Diffusion-based generation conditioned on edges obtained from ImageNet images ("ImageNet edges").** Instead of using edges produced from our 3D visual prompt module, the images are generated with the same diffusion model but conditioned on edges computed from ImageNet images with the Canny edge detector (Canny, 1986).

**Rendered images with random 2D image backgrounds ("rendering + BG2D").** We overlay the rendered objects to a randomly sampled background image. The background images come from the training split of BG-20k (Li et al., 2022) dataset with no salient objects.

**Rendered iamges with random 3D environment backgrounds ("rendering + BG3D").** We collected 100 HDRIs from polyhaven and used them as the world environment in Blender rendering. The 100 HDRIs cover a wide range of scenes, including both indoor and outdoor, natural and urban, and different lighting conditions (day or night).

## E    EXPERIMENTAL RESULTS ON 3D OBJECT DETECTION

To show the flexibility of our 3D-DST approach and its potential for a wide range of 3D tasks, we present qualitative and quantitative results on 3D object detection from 3D scenes on ARKitScenes dataset (Baruch et al., 2021; Brazil et al., 2023). In the following, we start by presenting our DST-3D data generation for 3D scenes with multiple objects. Then we visualize qualitative examples in Figure 9. Finally we report the quantitative results in Table 7. **Results show that (i) our DST-3D approach can generate synthetic scene images with multiple objects from different categories, and (ii) state-of-the-art models pre-trained on DST-3D data can achieve improved performance for 3D object detection in terms of both AP2D and AP3D.**

**DST-3D data generation for 3D scenes.** Unlike 2D image classification or 3D pose estimation, generating 3D scene images for 3D object detection consists of the generation of multiple objects from multiple categories in a reasonable 3D scene structure. Therefore, we adopt the scene structures from the ARKitScenes dataset (Baruch et al., 2021) and extend our 3D visual prompt module (see Figure 2) as follows. In ARKitScenes, each 3D scene is represented by a list of 3D bounding boxes, each associated with a category label. We start by sampling a CAD model from the target category and putting it in the Blender scene at the target location and with the target rotation. The synthetic 3D scene possesses a reasonable scene structure (object location and rotation) with randomness inherent in the CAD models (object styles and shape ratios). Finally, we sample different camera locations and rotations to generate a series of images of the same scene but from different viewpoints.

**Qualitative examples of 3D synthetic scenes generate with our DST-3D.** In Figure 9 we visualize several qualitative examples of scene images produced by our DST-3D. We can see that our DST-3D approach can generate scene images with multiple objects from multiple categories with diverse viewpoints.

**Quantitative results of 3D object detection on ARKitScenes dataset.** We choose CubeR-CNN (Brazil et al., 2023) from CVPR 2023 as the baseline model and report the 3D object detection performance on ARKitScenes dataset (Baruch et al., 2021) in Table 7. Results show that by pretraining 3D object detection models on synthetic DST-3D scene images, we achieve improved performance in terms of both AP2D and AP3D metrics.

## F    ANALYZING THE QUALITY OF THE AUTOMATIC 3D ANNOTATIONS

In Table 4 and Section E, we showed that improved performance on both in-distribution data and out-of-distribution data can be achieved by pretraining 3D models on our DST-3D data. However, it remains unclear how much of the synthetic data has accurate 3D annotations or how much noise was introduced from our DST-3D data generation.

In this section, we first collect feedback from human evaluators and analyze the failure cases of our generation pipeline. Then, we introduce two automatic metrics to analyze the quality of the automatic 3D annotations in our DST-3D data. Lastly, we propose a K-fold consistency filter (KCF) built on a state-of-the-art pose estimation method that can effectively filter noisy images with inaccurate 3D annotations from our DST-3D data, hence improving the overall quality. We also present quantitative and qualitative results to support our argument.

**Human evaluations.** We collect feedback from human evaluators to analyze the quality of our generated dataset. Specifically, we developed a Gradio app and presented the human evaluator with the textual prompt, the visual prompt, and our 3D-DST image. Then, for each triplet presented, the human evaluator would determine if the 3D-DST image is consistent with the textual and visual prompts. We visualize the collected results in Figure 5 and find that around 75% of our 3D-DST images are consistent with both the textual and visual prompts.

Moreover, by analyzing the failure samples, we identify a limitation of our model to be images with challenging and uncommon viewpoints, such as looking at cars from below or guitars from the side. Some failure samples are demonstrated in Figure 10.

**Measuring the quality of 3D annotations.** Two metrics were considered to measure the quality of 3D annotations in a synthetic 3D dataset.

1. **Real-to-synthetic performance.** Given a powerful 3D pose estimation model (Ma et al., 2022) trained on real data, we evaluate the model on the synthetic dataset. A good real-to-synthetic performance indicates the consistencies between the real data and the synthetic data, as well as the general quality of the 3D annotations in the synthetic dataset. However, a critical limitation of this metric is the domain gap between the synthetic and real. For instance, a synthetic dataset with higher diversity than real data, e.g., our DST-3D data with diverse LLMs, could lead to a lower real-to-synthetic performance due to unseen textures, weather, poses, etc.

2. **Pose consistency score (PCS).** By randomly splitting the synthetic data into training sets and validation sets, the *pose consistency score (PCS)* is given by a model's average performance on the validation sets after fitting the corresponding training sets. Since the training data and validation data share the same source domain, synthetic data with noisy 3D labels would yield a lower PCS as the model cannot recover the noises added from the image generation module. It should be noted that the absolute value of PCS is also affected by the fitting capabilities of existing 3D models and the diversity of the synthetic data.

**K-fold consistency filter (KCF).** We propose a simple yet effective approach to filter out noisy samples with inaccurate 3D annotations. We follow the K-fold splits and for each train-val split, we train a state-of-the-art 3D pose estimation model (Ma et al., 2022) and evaluate it on the validation split. For each sample in the validation split, we regard it as a noisy sample with possibly inaccurate 3D annotation if the confidence score given the 3D annotation is lower than a threshold.

**Quantitative and qualitative examples of KCF.** To show the efficacy of the K-fold consistency filter (KCF), we present quantitative results on four categories: *bus*, *car*, and *microwave*. Results in Table 8 show that KCF can effectively remove the noisy samples with inaccurate 3D annotations with increases in both R2S and PCS. We further visualize samples with high confidence scores (possibly accurate) and low confidence scores (possibly noisy) in Figure 10. We find that KCF is working as expected, and most samples removed by KCF are indeed noisy and have inaccurate 3D annotations. However, we observe few samples mistakenly filtered by KCF. We conjecture this is due to the diverse image space explored by DST-3D and the limited robustness of existing 3D pose estimation methods, on which KCF is built.

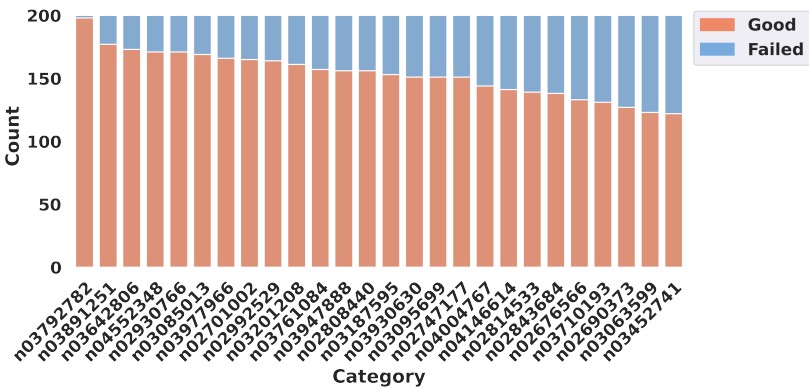

Figure 5: **Human evaluation results.** We collect feedback from human evaluators and find that about 75% of the generated 3D-DST images are consistent with both the textual and visual prompts. By analyzing the failure cases, we identify a limitation of our model to be images with challenging and uncommon viewpoints.

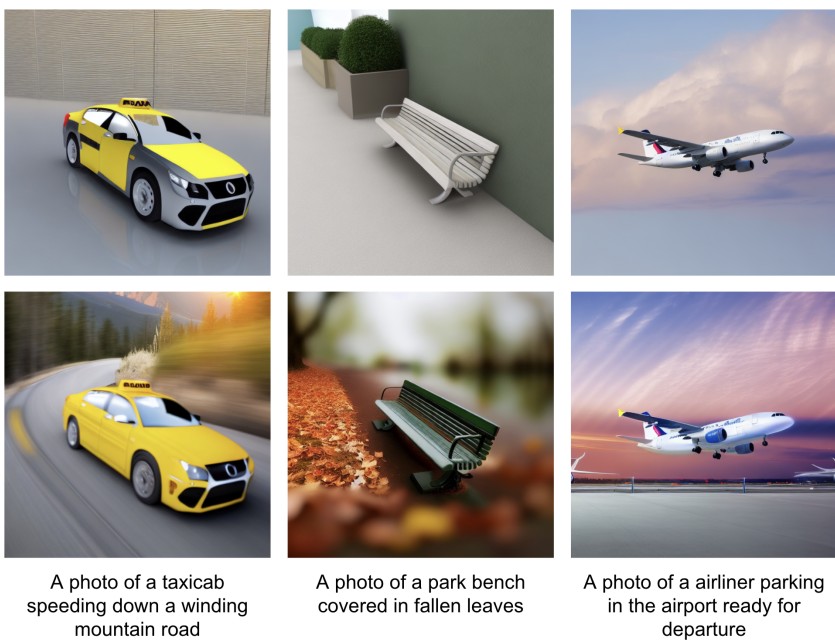

| A photo of a taxicab speeding down a winding mountain road | A photo of a park bench covered in fallen leaves | A photo of a airliner parking in the airport ready for departure |

Figure 6: **Qualitative examples demonstrating the effectiveness of LLM prompts.** We conduct ablation study experiments by comparing the quality of the synthetic images obtained from the same 3D visual prompts with (top row) or without (bottom row) LLM prompts. We show that LLM prompts provide extra information about the appearance of the foreground object or the background scene and can effectively improve the diversity and quality of synthetic images.

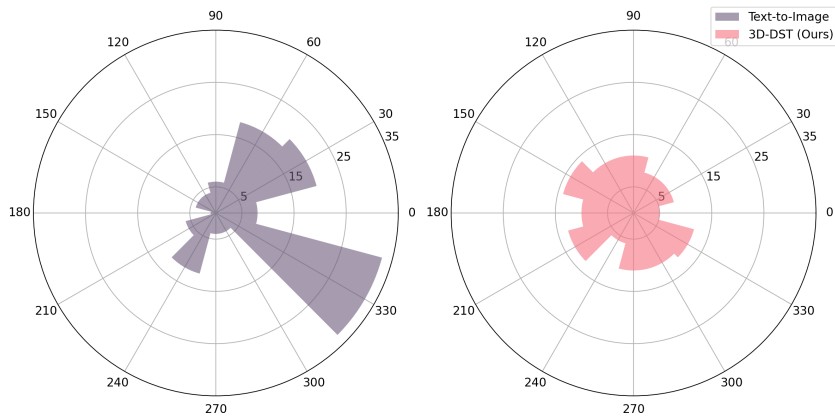

Figure 7: **Distribution of azimuth angles of objects in Text2Img (left) and our 3D-DST (right).** We visualize the distribution of the azimuth angles for 120 objects in Text2Img (left) and 3D-DST (right). Synthetic data in Text2Img show strong pose biases propagated from the image diffusion model, while our 3D-DST dataset alleviates this issue by introducing explicit 3D controls.

| Azimuth | Elevation | Theta | Classes |
|---------|-----------|-------|---------|
| all | all | $\mathcal{N}(0, \pi/18)$ | airliner, beach wagon, cab, coffee mug, dining table, piano bicycle, pillow, police van, pot, school bus, warplane, bottle bench, birdhouse, ambulance, trolleybus |
| front | all | $\mathcal{N}(0, \pi/18)$ | cellular phone, laptop, mailbox, microwave, remote control washer, bag |
| all | top | $\mathcal{N}(0, \pi/18)$ | keyboard, table lamp, trash can, bathtub, couch, soup bowl |
| front | top | $\mathcal{N}(0, \pi/18)$ | printer, stove |

Table 5: Viewpoint sampling rules for 3D-DST generated for image classification.

| Methods | In-distribution (ID) | | Out-of-distribution (OOD) | |
|---------|--------|-----------|--------|-----------|
| | DeiT-S | ConvNeXt-S | DeiT-S | ConvNeXt-S |
| Baseline | 84.56 | 91.34 | 49.61 | 67.19 |
| *w /* Text2Img (2023) | 84.80 (↑0.24) | 91.26 (↓0.08) | 50.83 (↑1.22) | 67.50 (↑0.31) |
| *w /* ImageNet edges | 85.20 (↑0.64) | 91.08 (↓0.26) | 48.69 (↓0.92) | 69.29 (↑2.10) |
| *w /* Rendering + BG2D | 84.96 (↑0.40) | 91.50 (↑0.16) | 50.92 (↑1.31) | 68.42 (↑1.23) |
| *w /* Rendering + BG3D | 87.50 (↑2.94) | 91.66 (↑0.32) | 54.81 (↑5.20) | 68.17 (↑0.98) |
| *w /* 3D-DST (ours) | **88.96** (↑4.40) | **92.18** (↑0.84) | **56.65** (↑7.04) | **69.69** (↑2.50) |

Table 6: **Ablation study of data generation methods on image classification.** We report top-1 accuracy (%) on ImageNet-100 (ID) and ImageNet-R (OOD) using DeiT-S and ConvNeXt-S. We compare the performances when models are trained purely on the target dataset or pre-trained on synthetic dataset with different data generation methods. Please refer to Section D for specifics of each data generation approach.

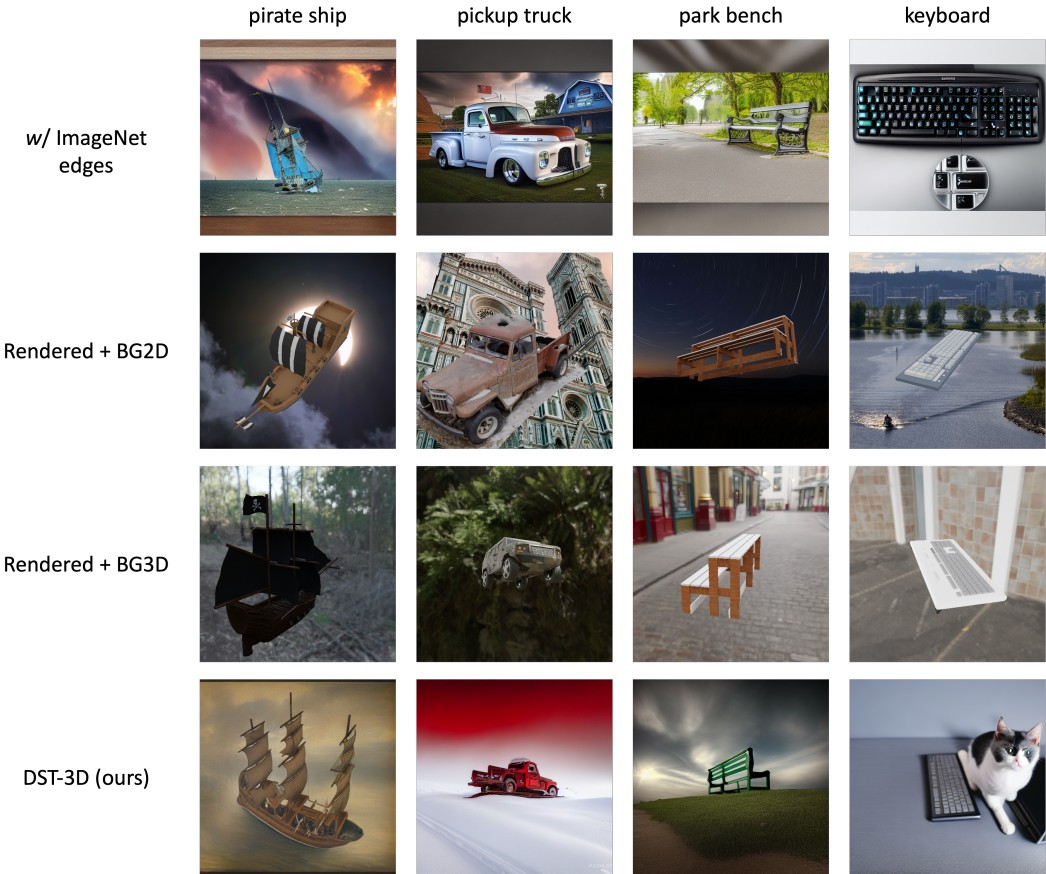

Figure 8: Qualitative examples of different data generation methods for ablation study. Please refer to Section D for specifics of different methods.

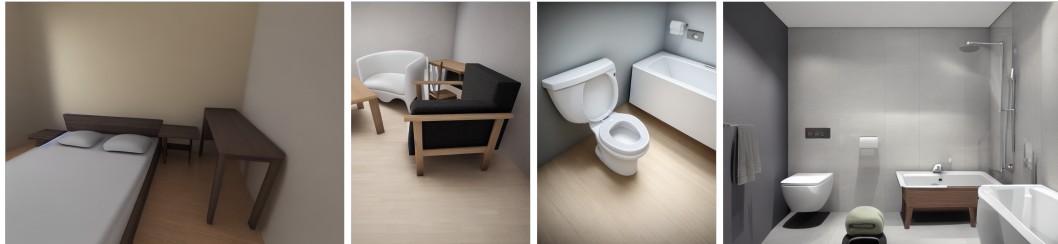

Figure 9: **Qualitative examples of scene images produced by our DST-3D.** We extend our 3D visual prompt module with 3D scene structures from ARKitScenes dataset (Baruch et al., 2021). We can see that our DST-3D can produce synthetic images with multiple objects from multiple categories with diverse viewpoints. State-of-the-art 3D object detection models pretrained on these scene images can achieve improved performance (see Section E).

| Methods | AP2D | AP3D |
|---|---|---|
| CubeRCNN (Brazil et al., 2023) | 41.50 | 41.65 |
| *w /* DST-3D (ours) | **42.34** (↑0.84) | **42.74** (↑1.09) |
| *w /* DST-3D + camera aug (ours) | **42.86** (↑1.36) | **43.19** (↑1.54) |

Table 7: **Quantitative results of 3D object detection on ARKitScenes dataset (Baruch et al., 2021).** We adopt the state-of-the-art 3D object detection model CubeRCNN (Brazil et al., 2023) from CVPR 2023 as the baseline model and compare the performance with or without DST-3D pretraining. Results show that by pretraining 3D object detection models on synthetic DST-3D scene images, we achieve improved performance in terms of both AP2D and AP3D metrics.

| Data | R2S (%) | PCS (%) |
|---|---|---|
| DST-3D (bus) | 65.8 | 59.5 |
| *w /* KCF | 67.1 (↑1.3) | 66.0 (↑6.5) |
| DST-3D (car) | 83.2 | 36.3 |
| *w /* KCF | 83.9 (↑0.7) | 43.8 (↑7.5) |
| DST-3D (microwave) | 87.6 | 61.3 |
| *w /* KCF | 88.1 (↑0.5) | 66.5 (↑5.2) |

Table 8: **Analyzing the quality of 3D annotations with real-to-synthetic performance (R2S) and pose consistency score (PCS).** We show that with KCF we can effectively filter noisy samples with inaccurate 3D annotations from our DST-3D data, hence improving the overall 3D annotation quality. Please refer to Section F for more details.

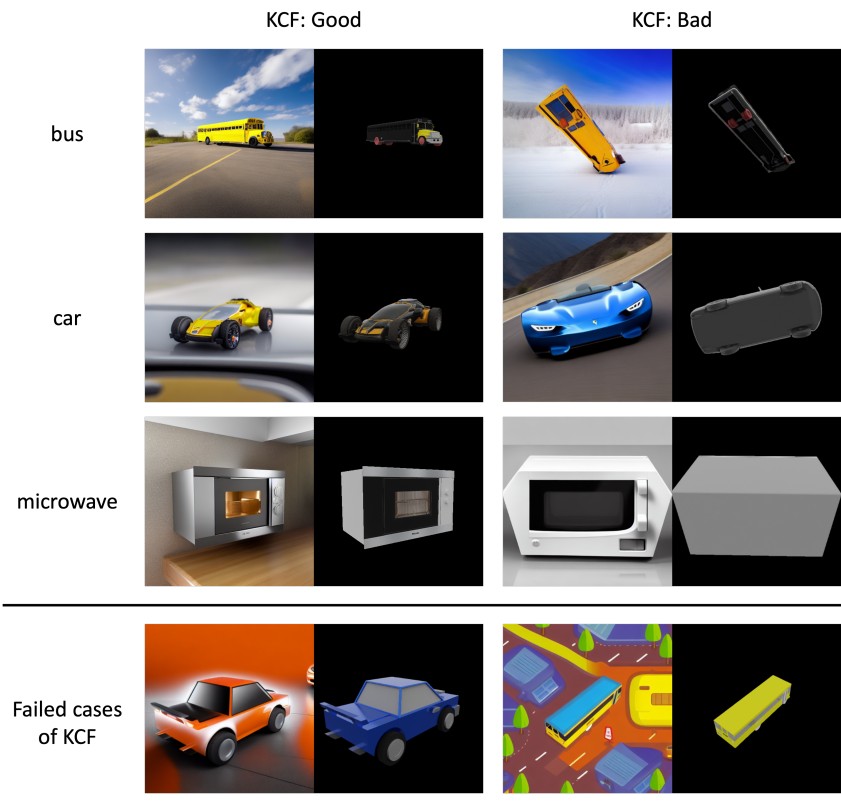

Figure 10: **Qualitative examples of K-fold consistency filter (KCF).** For each class we visualize samples predicted as good or bad by KCF. We can see KCF effectively filter out noisy samples with inaccurate 3D annotations. However, we also observe cases where KCF filters our images with accurate 3D annotation (bottom row). We conjecture this is due to the diverse image space explored by DST-3D and the limited robustness of existing 3D pose estimation methods, on which KCF is built.

