# OpenReview forum: "Generating Images with 3D Annotations Using Diffusion Models"
_ICLR.cc/2024/Conference — ICLR 2024 spotlight_

### Official Review · Reviewer_eSpe · 2023-10-29

**Soundness:** 3 good
**Presentation:** 3 good
**Contribution:** 2 fair
**Rating:** 5
**Confidence:** 3

**Summary:**

This paper investigate how to use generated images from diffusion models to improve discrimitive tasks in both in-distribution and out-of-distribution settings. The authors propose to render 3D data to 2D edge maps, fine-tune the large-scale diffusion model via ControlNet approach with the prompt augmented form LLM. After training, the generated images naturally have all the 3D object annotations. These generated data can be subsequently used as data augmentation for downstream tasks. The paper demonstrate the effectiveness of the proposed method on image classification and 3D pose estimation tasks.

**Strengths:**

- The proposed framework is simple and straightforward to use. The main technical contribution seems to be how to better use 3D data, ControlNet and LLM for data augmentation.
- Quantitative improvement looks promising. On the evaluated task (image classification and pose estimation), the quantitative improvement seems quite obvious, espcially for the OOD settings.

**Weaknesses:**

- The main method aims to produce (image, 3D annotation) pairs. Then is 2D image classification a good task for evaluation? The corresponding 3D ground truth is not used anywhere in this task. And you don't need 3D data to create (image, label) pairs. Even though experiments indeed show the improvement, I doubt this could be achieved without using 3D data.
- As the main purpose is to use the generated data for downstream tasks, I think the paper needs to carefully examine the data quality and show the necessity of the proposed approach. From this aspective, some necessary ablations are missing.
  - One simplest baseline to use 3D data is to just use the rendered image with background (e.g., random environment map). Would this kind of synthetic data improve the evaluated tasks? I think this baseline is needed to prove the necessity of using a image generative model.
  - For image classification, a simpler approach is to just use the imagenet images, extract the edge map and then generate new image conditioned on the LLM prompt. Would this kind of synthetic data also give big improvement? This baseline is needed to show the necessity of using the 3D data, as least for the image classification task.
- The title is a bit misleading. It seems to suggest a method that enable 3D control of the diffusion model (e.g., changing view points), but it's not. The proposed method merely use the 3D data and diffusion model to create (image, 3D annotation) pairs.

**Questions:**

What is the prompt to LLM for enriching the description?

---

> ### Author Response · Authors · 2023-11-20
> **Author's Response to Concerns Raised by Reviewer eSpe**
>
> We thank Reviewer eSpe for the valuable feedback and constructive comments. Here are our responses to each concern.
>
> &emsp;
>
> **`W1` Purpose of 2D image classification task**:
>
> In this work, we present a synthetic data generation approach, 3D-DST, that produces synthetic images with controllable viewpoints and automatic 3D annotations at no extra cost. To demonstrate the advantage of our approach, we provide the following experimental results:
> 1. Improved 2D image classification on both in-distribution and out-of-distribution datasets (Table 1) by using our 3D-DST data for pretraining. These results demonstrate that our synthetic images possess a high level of realism (for ID classification) and diversity (for OOD classification) compared to other synthetic data, such as the 2D images in Text2Img.
> 2. Improved 3D pose estimation (Table 4) and improved 3D object detection (General Response 3, Section E). These results not only show the great potential of our approach on multiple 3D tasks but also verify the quality of the automatically obtained 3D annotations in 3D-DST.
>
> We agree with the reviewer that improved 2D image classification can be achieved with standard 2D diffusion models without 3D-DST. In Table 3 we compared our approach with Text2Img that utilizes a 2D diffusion model for synthetic image generation, along with other complex modules such as language enhancement and CLIP filtering. Results show that pretraining on our 3D-DST images outperforms pretraining on Text2Img by a wide margin for both ID and OOD settings.
>
> Despite the fact that 2D image classification doesn't rely on the automatic 3D annotations generated by 3D-DST, we argue that our approach is still a compelling choice for 2D tasks. The 3D visual prompts effectively ensure the class of the salient object, and with our 3D control module, viewpoints of objects in our 3D-DST images follow a uniform distribution, as opposed to the strong biases observed from a standard image diffusion model (Figure 6).
>
>
> &emsp;
>
> **`W2` Some necessary ablations are missing**:
>
> We thank the reviewer for pointing out the missing ablation experiments. Please refer to **General Response 4** where we present the new ablation study results.
>
> &emsp;
>
> **`W3` The title is a bit misleading**:
>
> Thanks for the suggestion. We are considering using **“Generating Images with 3D Annotations using Diffusion Models”** as the new title. If you have other suggestions for the title, we are very glad to modify it accordingly.
>
> &emsp;
>
> **`Q` What is the prompt to LLM for enriching the description?**
>
> In contrast to previous works (e.g., VisProg), which directly acquire desirable outputs from LLM-powered chatbots by feeding curated prompts describing the tasks and several example outputs for in-context learning, our approach leverages the generative power of LLMs to produce a series of diverse prompts describing the target objects. More specifically, we start from “A photo of a [class]”, and run LLM in an auto-regressive manner. We find that LLMs are capable of giving detailed descriptions regarding the color, texture, and shape information of the object, as well as diverse scene environments and actions. As demonstrated in Figure 5, prompts like “A photo of a taxicab covered in a thick layer of snow” and “A photo of a park bench covered in fallen leaves” give diversity to the objects’ appearance. And prompts like “A photo of a park bench nestled under the shade of a towering oak tree” enrich the background of the generated images. We will release our codebase with the implementation of our LLM generation module to reproduce our results.

---

> > ### Author Response · Authors · 2023-11-22
> > **Official comment by authors**
> >
> > Dear Reviewer eSpe,
> >
> > Thanks again for your valuable feedback and suggestions. We are wondering if our response has addressed your concerns. If there is any information or clarification you need from us, please feel free to let us know. We are happy to have further discussions!

---

> > ### Comment · Reviewer_eSpe · 2023-11-22
> > **Reply**
> >
> > Thanks for siginificant effort and detailed response from the authors.
> >
> > - The provided ablation experiments are great. I'm a bit confused when looking at Figure 7, in particular the "w imagenet edges" row. Why are the generated images have top and down horizontal frames, like the model is generating non-square images?
> >
> > - The new title is better than the old one.
> >
> > - After reading other reviews, I kind of agree that the technical contribution is a bit limited, i.e., the method itself seems to be a direct extension to ControlNet, even though I respect the authors' claim that the whole framework (method, datasets, experiments) provides insights to the community. I'm not a expert on image classification / pose estimation tasks, so I'm not sure how such quantitative improvement should be appreciated.
> >
> > I'm not fully convinced to raise my score at this moment, but I do feel more positive towards this paper.

---

> ### Author Response · Authors · 2023-11-22
> **Official comment by authors**
>
> Dear Reviewer eSpe,
>
> Thank you very much for the valuable feedback and constructive comments!
>
> &nbsp;
>
> `Regarding the qualitative examples in Figure 7:`
>
> The horizontal frames are due to the ImageNet image preprocessing. Images are first upsampled and then padded to the same shape. Images with higher resolutions improve the performance of the Stable Diffusion and running diffusion models in batch mode significantly improves the data generation speed given limited time for the rebuttal experiments. In general, we find the quality of the produced images is not impacted by the image boundaries. We will re-run the experiments again without the boundaries.
>
> &nbsp;
>
> Again, we appreciate the reviewer’s continuous efforts to provide helpful suggestions and valuable input, which make the paper better. If there is any further information or clarification you need from us, please feel free to let us know!

---

### Official Review · Reviewer_B6MH · 2023-10-30

**Soundness:** 2 fair
**Presentation:** 3 good
**Contribution:** 2 fair
**Rating:** 5
**Confidence:** 4

**Summary:**

- The author introduces a method named 3D-DST, aimed at enhancing the comprehension of 3D objects by diffusion models.
- This method comprises two modules: the "3D Visual Prompt" module, which utilizes edge maps as prompts derived from rendering, and the "Text Prompt" module, which extends prompt words through LLM.
- Through experiments, the author demonstrates that the images generated by the proposed method, along with paired labels, serve as an effective approach for data enhancement or pre-training. This leads to improved performance in tasks such as image classification and 3D pose estimation across multiple baselines.

**Strengths:**

- The paper is easy to understand.
- The paper presents an approach that incorporates edge maps as additional prompts to enhance the performance of the diffusion-based method.

**Weaknesses:**

- The framework is mainly inherited from Controlnet, so the technical contributions are limited and incremental.
- The idea of 3D Visual Prompt via CG rendering and LLM Prompt is more like a combination of multiple previous effective techniques.
- The author's excessive focus on introducing background knowledge of known technologies like diffusion or cross-attention is unnecessary if the method utilized in this article relies on off-the-shelf approaches. It is not recommended to extensively discuss these technologies in the main text.
- The second challenge, "simple text prompts," seems to be less directly relevant to the paper's introduction on adding 3D geometry control to diffusion models.
- The experimental part of the paper lacks details on training the network.

**Questions:**

How to define camera extrinsic matrix and whether to use class-level canonical-space as the extrinsic matrix of identity. If this is the case, there are many symmetric objects whose poses are ambiguous (this issue has been extensively discussed in the work on object 6dof estimation). How to define objects with multiple symmetry axes such as round tables? In addition, how to align the definition of extrinsic coordinate systems between different classes?
It is counterintuitive to claim that edge maps are superior to depth maps because depth maps provide more 3D information, such as occlusion relationships, which goes beyond the 2D representation of edge maps. The conclusions presented in the author's paper are difficult to support with only a few selectively chosen qualitative examples.

---

> ### Author Response · Authors · 2023-11-20
> **Author's Response to Concerns Raised by Reviewer B6MH (1/2)**
>
> We thank Reviewer B6MH for the valuable feedback and constructive comments, and we address the weaknesses below.
>
> &emsp;
>
> **`W1` Limited technical contributions**:
>
> Thanks to the reviewer for the comments. Please refer to **General Response 2** where we summarize our results and highlight our contributions.
>
> &emsp;
>
> **`W2&3` The idea of 3D Visual Prompt via CG rendering and LLM Prompt is more like a combination of multiple previous effective techniques. The author's excessive focus on introducing background knowledge of known technologies like diffusion or cross-attention is unnecessary if the method utilized in this article relies on off-the-shelf approaches**:
>
> We respectfully disagree with the reviewer’s biased opinions against works built on existing methods. A successful work builds on existing technologies but solves new and challenging problems [A]. In this work, we show that with the help of diverse LLM prompts and controllable visual prompts, we can obtain high-quality images with 3D annotations. We demonstrate that using 3D-DST data for pretraining can achieve improved performance for image classification, 3D pose estimation, and 3D object detection. By combining graphics-based rendering with strong generative power of diffusion models and our diverse LLM prompts, we achieve promising results on both in-distribution data and out-of-distribution data.
>
> In Section 3.1 we discussed the background of existing generative methods (diffusion models and ControlNet) for readers lack of related context, which is a common practice. Moreover, we discussed the limitations of existing methods and presented the motivation of our work, as a way to highlight our contributions. Given the reviewer’s advice, we will re-organize Section 3.1 to make the contents more succinct and highlight our key ideas.
>
> [A] T Gupta et al. Visual programming: Compositional visual reasoning without training. In CVPR, 2023.
>
>
> &emsp;
>
> **`W4` Simple text prompts seem to be less directly relevant to the paper's introduction on adding 3D geometry control to diffusion models**:
>
> We acknowledge that the current introduction in the manuscript was not very clear about the challenges with “simple text prompts” and the motivation of our “diverse text generation”. We have revised this in the next revision. Here we would like to make two arguments motivating the necessity of LLM prompts.
>
> 1. Unlike typical 2D diffusion models such as Stable Diffusion, introducing 3D visual prompts as a condition to diffusion models largely limits the diversity of background and appearances, as well as the richness of details. This motivates us to encourage the diversity of output images from textual inputs, leading our proposed approach of generating diverse text prompts with LLMs. We qualitatively compare the generated images with and without LLM prompts in Figure 5.
>
> 2. In terms of the purpose of our work, we aim to develop a synthetic data generation method with 3D annotations that demonstrates better realism and diversity compared to graphics-based rendered data. Thus diverse LLM prompt serves as a core module of our approach and is key to achieving improved performance on OOD data in downstream tasks as presented in Table 3.
>
> &emsp;
>
> **`W5` Training details of networks.**:
>
> Given the limited space in the main text, we move some of the model training specifics from Section 4.1 “Implementation details'' to Section B in the supplementary materials. In Section 4.1 we report the network architectures, model sizes, and synthetic data size. In Section B of supplementary materials, we go through the choice of hyperparameters and training settings for both image classification and 3D pose estimation. Hyperparameters not discussed in our manuscript follow the choices from the publicly released code of DeiT, ConvNeXt, etc. We are happy to improve the reproducibility of our experiments so please let us know if additional details should be covered in our revision.

---

> ### Author Response · Authors · 2023-11-20
> **Author's Response to Concerns Raised by Reviewer B6MH (2/2)**
>
> We answer the questions raised by reviewer B6MH below.
>
> &emsp;
>
> **`Q1` How to define camera extrinsic matrix and whether to use class-level canonical-space as the extrinsic matrix of identity. How to define objects with multiple symmetry axes such as round tables?**
>
> Our 3D-DST generation pipeline saves pose annotations in terms of the 3D viewpoints (azimuth, elevation, and in-plane rotation), as well as object distance, object center, and Blender camera parameters. This follows the convention in standard pose estimation datasets, such as PASCAL3D+ and ObjectNet3D. Similar to other 3D datasets such as Omni3D, the intrinsic matrix can be obtained from the focal length and principal point of the camera, and the extrinsic matrix is simply an identity matrix as we use a canonical camera. In 3D-DST, the object rotation matrix can be computed from the viewpoint parameters.
>
> Moreover, as all CAD models share the same annotation convention, we further ensure the T-pose of all CAD models are aligned both in-category and across-category. For instance, the “front” of a vehicle and the “front” of a dog would face the same direction as they share similar semantic parts. This allows 3D models trained on our synthetic 3D-DST data to exploit semantic similarities across categories.
>
> Regarding how to define objects with multiple symmetry axes such as round tables, we simply follow the convention in PASCAL3D+ and ObjectNet3D and set the azimuth rotation to zero. This applies to both rigid objects commonly used for pose estimation, such as round tables, bowls, and bottles, as well as other objects less used, such as apples and golf balls. As mentioned by the reviewer, these symmetry issues have been well-studied by previous works in pose estimation. In this work, we simply follow the convention in downstream tasks and we find models pretrained on our 3D-DST data working fairly well for both 3D pose estimation (Table 4) and 3D object detection (General Response 3, Section E).
>
> &emsp;
>
> **`Q2` It is counterintuitive to claim that edge maps are superior to depth maps because depth maps provide more 3D information, such as occlusion relationships, which goes beyond the 2D representation of edge maps.**
>
> We acknowledge that further study is necessary regarding the choice of edge maps or depth maps as the 3D visual prompts. We make the following observations:
>
> 1. The current inferior results when using depth maps as 3D visual prompts are attributed to the domain gap between the MiDaS depths used for training and the depths used for inference. Running MiDaS on synthetic images or running diffusion models conditioned on synthetic depths both result in large domain gaps with the MiDaS depth maps obtained from real images during training. We believe certain finetuning is necessary to close the domain gap and to fully demonstrate the advantage of depth maps over edge maps. We leave this as an interesting direction for future work.
> 2. Although edge maps provide less 3D information than depth maps, it is still a compelling choice for 2D image generation despite its simplicity. With the scale-distance ambiguity, it abstracts away the need to sample a wide range of scales and distances during rendering, and directly provides necessary shape information that has been “projected” onto the 2D plane. Moreover, results on scene image generation with 3D-DST (in Section E of supplementary materials) also show that edge maps work fairly well for multiple objects with mutual occlusion.

---

> > ### Comment · Reviewer_B6MH · 2023-11-22
> >
> > Dear authors,
> >
> > Thank you for your response.
> >
> > I have read all the reviewers' comments and the authors' rebuttal and some of my concerns are addressed, e.g. the 3D pose annotations. However, the authors' responses to W4 and Q2 are not entirely convincing to me. Regarding the technical contribution and novelty of the paper, I respect the authors' perspective on 'building upon existing technology' in their response to W2 & W3. However, I want to point out that the contribution summarized from the current structure of the paper (i.e., the 3D-DST framework) seems more like an extension of ControlNet rather than a complete work.
> >
> > I appreciate the effort made by the authors in the rebuttal, but I believe that for the paper to be more self-consistent, further improvements and supplements are needed in the writing and experimental sections. For example, addressing the misleading aspects of the paper title mentioned by other reviewers, or their concerns about consistency.
> >
> > Even though I do not consider the current state of the paper to reach an acceptable standard, I will also consider the opinions of other reviewers to decide the final score.

---

> > > ### Author Response · Authors · 2023-11-22
> > > **Official comment by authors**
> > >
> > > Dear Reviewer B6MH,
> > >
> > > Thank you so much for your further comments! We really appreciate it!
> > >
> > > &nbsp;
> > >
> > > `About the contributions:`
> > >
> > > We appreciate and respect the reviewer’s opinion. Here, we hope to highlight the differences between our 3D-DST and ControlNet:
> > >
> > > -  The core contribution of our method is a simple yet effective framework to create diverse 2D images with 3D annotations, while the original ControlNet focuses on 2D only. This data generation idea is not a simple extension from ControlNet, but also effective methods and experiments to validate the quality of the 3D annotations, improve the diversity of the generated images, and apply the 3D-DST to various downstream tasks and improve the SOTA performance.
> > > - We propose a method leveraging LLMs to generate prompts, which help us to create more diverse samples. This also enables us to improve out-of-distribution (OOD) performance, e.g., on ImageNet-R, by pretraining on our 3D-DST data. This is unexplored in previous works.
> > > - We conduct extensive experiments using our generated dataset on both ID and OOD data for image classification, 3D pose estimation, and 3D object detection. These experiments focus on very different problems from the original ControlNet.
> > >
> > > Even though our framework is built on ControlNet, we believe our method, dataset, and experiments provide new insights and results for synthetic data generation, and can be useful resources for the community, especially on tasks where 3D annotations are difficult to obtain. Therefore, we sincerely hope the reviewer could reconsider our contributions.
> > >
> > > &nbsp;
> > >
> > > `Further improvements and supplements:`
> > >
> > > Thanks for pointing this out. We would really appreciate it if the reviewer could provide some more specific suggestions of what should be improved or updated. We will try our best to make the modifications in the revision or in the final version.
> > >
> > > &nbsp;
> > >
> > > `Paper title:`
> > >
> > > We are considering using “Generating Images with 3D Annotations using Diffusion Models” as the new title. If you have other suggestions for the title, we are very glad to modify it accordingly.
> > >
> > > &nbsp;
> > >
> > > Thanks again for the insightful comments from the reviewer. We’re looking forward to more specific suggestions, and we are very happy to update our manuscript according to them. Besides, we sincerely hope the reviewer can reconsider our contributions.

---

### Official Review · Reviewer_5ugh · 2023-10-31

**Soundness:** 3 good
**Presentation:** 3 good
**Contribution:** 2 fair
**Rating:** 6
**Confidence:** 4

**Summary:**

A method to add 3D geometry control into the image generation process. To achieve the goal, three key techniques are leveraged, including 2D edge maps generation with 3D annotations via rendering, text prompt generation for improving the diversity, and conditional image generation from edge maps and text descriptions. The method can be utilized as a data augmentation strategy for many downstream tasks such as image classification. Experiments can show its promising application potential.

**Strengths:**

- Adding 3D geometry control via 2D edge maps and text descriptions is interesting and reasonable. This way the generative model only needs to deal with controlling information represented in 2D images and texts. Then many existing powerful techniques can be leveraged for controllable generation.
- The proposed method is reasonable. It can achieve plausible controllable and diverse generation results. Generated images are of good quality and well-related to edge prompts and text conditions.
- The method can serve as a promising data augmentation strategy for many downstream tasks. It is a promising way to generate diverse 2D images with 3D information annotation.

**Weaknesses:**

- The technical significance is relatively limited. The problem of generating 2D edge maps from 3D models and generating text prompts from 3D CAD models can be solved by existing techniques. Though the idea is interesting, no new techniques are proposed. The overall method is rather like an application-guided strategy. Though with promising application potential, it is hard to say what general principles that can guide the research in other domains can be distilled from the paper.
- It is not sure whether the generated images are very faithful to the edge maps conditions. For example, there is no good guarantee that the objects in the generated images are consistent with the geometry described via the edge maps.

**Questions:**

- Evaluations on whether the generated images are faithful to images and text conditions.
- It would be better if more potential applications could be discussed.

---

> ### Author Response · Authors · 2023-11-20
> **Response to Reviewer 5ugh**
>
> We thank the reviewer for the valuable feedback and constructive comments. Here are our responses to each question.
>
> &nbsp;
>
> **`[Q1]` Limited technical significance.**
>
> Thanks to the reviewer for the comments. Please refer to General Response 2 where we summarize our results and highlight our contributions.
>
> &nbsp;
>
> **`[Q2]` Though with promising application potential, it is hard to say what general principles that can guide the research in other domains can be distilled from the paper.**
>
> In general, our 3D-DST presents the following conclusions and guidelines that may benefit other 3D-from-2D tasks:
>
> 1. By leveraging the strong generative power of diffusion models, our 3D-DST data generation approach is an effective way to produce images with 3D annotations that demonstrate a higher level of realism and diversity. We demonstrate improved performance on both indoor and outdoor scenes with single-object or multiple-object images. These results show that 3D-DST is a generic approach with great potential in a wide range of 3D-from-2D tasks.
> 2. Despite the availability of 3D annotations from synthetic data, previous synthetic data generation methods were often hindered by the domain gap between synthetic and real, and the huge efforts to produce complex textures, realistic materials, and complex lighting conditions in graphics-based rendering. 3D-DST addresses the above problems with the powerful generative capabilities of diffusion models and allows us to quickly generate synthetic images for various 3D-from-2D tasks. In this work, we only consider existing 3D annotations from real images, but a lot of more useful 3D annotations as by-products can be exploited in future works.
> 3. Using graphics-rendered images is widely exploited in previous works as an effective approach to improve models’ performance on standard benchmarks. However, with 3D-DST, we argue that generating diverse synthetic images for pretraining is also an important approach to improve models’ robustness on OOD data, as demonstrated by 3D pose estimation on OOD data in Table 4. This is accomplished by our proposed 3D-DST pipeline and is not possible from previous synthetic data methods.
>
> &nbsp;
>
> **`[Q3]` There is no good guarantee that the objects in the generated images are consistent with the geometry described via the edge maps. Evaluations on whether the generated images are faithful to images and text conditions.**
>
> Thanks to the reviewer for raising this important question. We develop two metrics to measure the quality of the 3D annotations, namely real-to-synthetic performance (R2S) and pose consistency score (PCS). Moreover, we develop a simple yet effective approach, K-fold consistency filter (KCF) to remove images likely with wrong 3D annotations. We present some quantitative results and qualitative examples to demonstrate the quality of 3D annotations in our 3D-DST data. For future revision, we will conduct a thorough analysis and involve human comparison to quantitatively study the quality of 3D annotations in our 3D-DST data, as well as the efficacy of KCF. Please refer to General Response 1 and Section F for detailed results and discussions.
>
> &nbsp;
>
> **`[Q4]` It would be better if more potential applications could be discussed.**
>
> We agree with the reviewer that more 3D applications should be discussed. Therefore in General Response 3 and Section E in supplementary materials, we present the qualitative and quantitative results of 3D object detection with 3D-DST. Compared to other results shown in our paper, generating synthetic data for 3D object detection (i) involves scene image generation with multiple objects from multiple categories, and (ii) requires both 2D bounding box and 3D bounding box annotations to be accurate enough. Results show that our 3D-DST can also produce good scene images with multiple objects and diverse viewpoints, and state-of-the-art models pretrained on our 3D-DST scene images achieve improved performance on downstream tasks.

---

> > ### Author Response · Authors · 2023-11-22
> > **Official comment by authors**
> >
> > Dear Reviewer 5ugh,
> >
> > Thanks again for your valuable feedback and suggestions. We are wondering if our response has addressed your concerns. If there is any information or clarification you need from us, please feel free to let us know. We are happy to have further discussions!

---

> > > ### Comment · Reviewer_5ugh · 2023-11-23
> > >
> > > Dear authors,
> > >
> > > Thanks for your responses. While I still value the concept presented in the paper, I maintain my initial reservations regarding its limited contribution. I appreciate your dedicated efforts in addressing my concerns.

---

### Official Review · Reviewer_Xyyh · 2023-11-01

**Soundness:** 3 good
**Presentation:** 3 good
**Contribution:** 3 good
**Rating:** 8
**Confidence:** 4

**Summary:**

This paper presents a novel method for adding 3D geometry control to diffusion models such that recognition models pre-trained on diffusion models generated synthetic data and then trained on target datasets have performance gains on classic tasks like 2D image classification and 3D pose estimation.

**Strengths:**

- The idea in this paper is neat, simple yet effective.
- The idea is also very novel.
- The empirical improvements on ImageNet classification and pose estimations are solid, significant, and surprising.

**Weaknesses:**

- In table 4, why the baseline result NeMo w/ AugMix is missing?
- Could you discuss or ablate using other rendering types other than canny edges? Does canny edges work the best and why?
- There is no discussion for limitations.

**Questions:**

see weakness

---

> ### Author Response · Authors · 2023-11-20
> **Response to Reviewer Xyyh**
>
> We thank the reviewer for the valuable feedback and constructive comments. Here are our responses to each question.
>
> &nbsp;
>
> **`[Q1]` The baseline result NeMo w/ AugMix is missing.**
>
> The baseline result of NeMo w/ AugMix is missing due to limited time and computing resources when we prepare the manuscript. We have added this baseline experiment and updated the results in our revision (Table 4).
>
> &nbsp;
>
> **`[Q2]` Could you discuss or ablate using other rendering types other than canny edges? Does canny edges work the best and why?**
>
> In Section 4.3 and Figure 4, we visualized some examples and discussed the choice of different rendering types. In general, we find using edges as visual controls works the best. We acknowledge the fact that using depth maps could potentially provide useful 3D information for 2D image generation. However, this approach is mainly hindered by the domain gap between the synthetic depths (during inference) and MiDaS depths from real images (during training). Meanwhile, canny edges as visual prompts are a compelling choice despite its simplicity. With the scale-distance ambiguity, it abstracts away the need to sample a wide range of scales and distances during rendering, and directly provides necessary shape information that has been “projected” onto the 2D plane. Moreover, results on scene image generation with 3D-DST (in Section E of supplementary materials) also show that edge maps work fairly well for multiple objects with mutual occlusion.
>
> &nbsp;
>
> **`[Q3]` Missing discussion for limitations.**
>
> Thank the reviewer for pointing out this issue. We will involve a thorough analysis of limitations in our next revision. Here we briefly summarize our limitations:
>
> * **Limitations of 3D models:** Our data generation pipeline starts with the rendering of 3D object models. Despite the availability of large 3D model datasets, the abundance and diversity of 3D models can be limited for specific categories, such as gyromitra (a type of fungi) or plastic bags.
> * **Privacy and copyright issues:** As our model is built on large pretrained generative models, e.g., Stable Diffusion, the generated dataset may reflect data presented in the training set, violating privacy or copyright agreements.
> * **Biases in training data of generative model:** Although our model can generate diversified viewpoints and distances by adding 3D control to the data generation process, our pipeline may suffer from other biases inherent in the training data. As the Stable Diffusion model is trained on LAION-2B, known biases of the dataset may be observed in the 3D-DST dataset.

---

> > ### Author Response · Authors · 2023-11-22
> > **Official comment by authors**
> >
> > Dear Reviewer Xyyh,
> >
> > Thanks for the positive feedback and helpful suggestions! If there is any further information or clarification you need from us, please feel free to let us know.

---

### Author Response · Authors · 2023-11-20
**General response to all reviewers**

We thank the reviewers for finding our method “neat, simple yet effective” (Reviewer Xyyh), our approach “reasonable and promising” for many tasks (Reviewer 5ugh) and our results “solid and promising” (Reviewers Xyyh & eSpe). Below is our feedback on the general questions. More specific results are provided in the revision and highlighted in blue.

&nbsp;

**`[General Response 1]` Verifying the quality of 2D images and 3D annotations.**

In this work, we present the following results to demonstrate the quality of the generated 2D images and 3D annotations.

1. **2D Image classification on in-distribution data (ImageNet) and out-of-distribution data (ImageNet-R).** Results in Table 1 and General Response 5 show that classification models pretrained on our 3D-DST data effectively improve the classification performance on both ID and OOD data. These results show that images in our 3D-DST data contain salient objects from the target class with a high level of realism (for ID classification) and diversity (for OOD classification) compared to previous works.
2. **3D pose estimation (Table 4) and 3D object detection (General Response 3, Section E).** These results not only show the great potential of our 3D-DST on 3D tasks for both indoor and outdoor scenes, but also verify the quality of the automatically obtained 3D annotations.
3. **We present a simple yet effective approach, K-fold consistency filter (KCF), to further improve the overall 3D annotation quality (see Section F).** Results on real-to-synthetic performance (R2S) and pose consistency score (PCS) show that KCF can effectively filter samples with inaccurate 3D annotations. For future revision, we will conduct a thorough analysis and involve human comparison to quantitatively study the quality of 3D annotations in our 3D-DST data, as well as the efficacy of KCF.

&nbsp;

**`[General Response 2]` Regarding novelty and contributions.**

In this paper, we propose 3D-DST, a simple yet effective framework that allows us to seamlessly create a large-scale dataset with rich 3D annotations, such as 3D poses, key points, shapes, and 3D locations. The novel design of our 3D-DST offers us numerous advantages, as outlined below:

1. **Automatic generation of 3D annotations.** Our 3D-DST enables us to acquire 3D annotations for the generated images through the rendering process. Therefore, we are able to establish a large-scale dataset with rich 3D annotations. In contrast, existing generative models, such as Stable Diffusion and ControlNet, cannot give us 3D annotations.
2. **Generating images with multiple viewpoints.** With a controllable 3D visual prompt module, our method is able to generate images from desirable viewpoints, including those that are rarely encountered in typical scenarios. This not only helps to alleviate the viewpoint biases that commonly exist in image generative models (Figure 6), but as a synthetic pretraining dataset, also improves models’ robustness (Table 4). In contrast, existing generative models lack control over the viewpoint of the generated images, limiting their application in this regard.
3. **Diverse images from diverse LLM prompts.** In this work, we propose a novel strategy for text prompt generation using LLM. Hence our 3D-DST is capable of generating images with a high level of realism and diversity compared to previous synthetic datasets. This helps to (i) generate diverse images with low redundancy for efficient and effective pretraining, and (ii) improve models’ performance on both in-distribution and out-of-distribution data. Existing synthetic datasets lack this design and achieve improvements only on standard in-distribution data.

&nbsp;

**`[General Response 3]` Experimental results on 3D object detection.**

To show the flexibility of our approach and its potential for a wide range of 3D tasks, we present qualitative and quantitative results on 3D object detection from 3D scenes on the ARKitScenes dataset in Section E of the supplementary materials. Results show that (i) our approach can generate synthetic scene images with multiple objects from different categories with diverse viewpoints, and (ii) state-of-the-art models pretrained on 3D-DST data can achieve improved performance for 3D object detection in terms of both AP2D and AP3D.

&nbsp;

**`[General Response 4]` Additional ablation experiments.**

As suggested by Reviewer eSpe, multiple ablation experiments were added. Please refer to Section D in supplementary materials for specifics of each data generation method. Sample images from each of the following experiments are presented in Figure 7 of supplementary materials.

1. For image classification only, extracting edges from ImageNet images and generating new images conditioned on the extracted edges.
2. Without 3D-DST, directly using rendered images with random 2D image backgrounds for pretraining.
3. Without 3D-DST, directly using rendered images with random 3D environment backgrounds for pretraining.

---

### Meta-Review · Area_Chair_PxaC · 2023-12-12

**Metareview:**

The paper has received mixed recommendations, two acceptances and two rejections with rating 5, 6, 8, 5.

The main confusion of this paper stems from the original title: “Adding 3D Geometry Control to Diffusion Models”. Reviewers B6MH and eSpe expressed concerns that the proposed method does not improve the geometry control of diffusion models. The AC concurred with the assessment. However, this paper is actually about generating data using 3D assets and pre-trained diffusion models to improve visual recognition on various downstream 2D/3D tasks. In rebuttal, authors change the title to “Generating Images with 3D Annotations using Diffusion Models” which is much more faithful to the content of the paper and resolves some concerns from B6MH and eSpe. However, reviewers B6MH and eSpe still consider the technical novelty in viewpoint control to be limited.

After careful consideration of reviews and rebuttal, the AC concurred with reviewers Xyyh and 5ugh’s assessments that this paper proposes a good data augmentation method for visual recognition and addresses a common viewpoint bias in Internet data.

While the original title is misleading, the problem is fixed by renaming the paper title without the need of a major revision to the paper content. The contribution of using 3D synthetic data, diffusion models, and LLMs to improve visual recognition is clear. Consequently, the AC recommends accepting the paper.

**Justification For Why Not Higher Score:**

There are still concerns about quality of generated images and ablation study on different data generation methods. While these concerns have mostly addressed in the rebuttal and new supplementary materials, they need to be organized into the main manuscript to improve the quality.

**Justification For Why Not Lower Score:**

The paper introduces an interesting synthetic data generation method harassing 3D datasets, text-image diffusion, and LLMs for improving visual recognition capability. This is a promising direction for scaling data and offsetting the viewpoint bias in visual datasets scraped from Internet. It is worth sharing more broadly.

---

### Decision · Program_Chairs · 2024-01-16

Accept (spotlight)